# Pelee: A Real-Time Object Detection System on Mobile Devices

**Robert J. Wang, Xiang Li, Charles X. Ling** *
Department of Computer Science
University of Western Ontario
London, Ontario, Canada, N6A 3K7
{jwan563,lxiang2,charles.ling}@uwo.ca

## Abstract

An increasing need of running Convolutional Neural Network (CNN) models on mobile devices with limited computing power and memory resource encourages studies on efficient model design. A number of efficient architectures have been proposed in recent years, for example, MobileNet, ShuffleNet, and MobileNetV2. However, all these models are heavily dependent on depthwise separable convolution which lacks efficient implementation in most deep learning frameworks. In this study, we propose an efficient architecture named PeleeNet, which is built with conventional convolution instead. On ImageNet ILSVRC 2012 dataset, our proposed PeleeNet achieves a higher accuracy and over 1.8 times faster speed than MobileNet and MobileNetV2 on NVIDIA TX2. Meanwhile, PeleeNet is only 66% of the model size of MobileNet. We then propose a real-time object detection system by combining PeleeNet with Single Shot MultiBox Detector (SSD) method and optimizing the architecture for fast speed. Our proposed detection system[2], named Pelee, achieves 76.4% mAP (mean average precision) on PASCAL VOC2007 and 22.4 mAP on MS COCO dataset at the speed of 23.6 FPS on iPhone 8 and 125 FPS on NVIDIA TX2. The result on COCO outperforms YOLOv2 in consideration of a higher precision, 13.6 times lower computational cost and 11.3 times smaller model size.

## 1 Introduction

There has been a rising interest in running high-quality CNN models under strict constraints on memory and computational budget. Many innovative architectures, such as MobileNets [1], ShuffleNet [2], NASNet-A [3], MobileNetV2 [4], have been proposed in recent years. However, all these architectures are heavily dependent on depthwise separable convolution [5] which lacks efficient implementation. Meanwhile, there are few studies that combine efficient models with fast object detection algorithms [6]. This research tries to explore the design of an efficient CNN architecture for both image classification tasks and object detection tasks. It has made a number of major contributions listed as follows:

**We propose a variant of DenseNet [7] architecture called PeleeNet for mobile devices.** PeleeNet follows the connectivity pattern and some of key design principals of DenseNet. It is also designed to meet strict constraints on memory and computational budget. Experimental results on Stanford Dogs [8] dataset show that our proposed PeleeNet is higher in accuracy than the one built with the original DenseNet architecture by 5.05% and higher than MobileNet [1] by 6.53%. PeleeNet achieves a compelling result on ImageNet ILSVRC 2012 [9] as well. The top-1 accuracy of PeleeNet is 72.1%

which is higher than that of MobileNet by 1.6%. It is also important to point out that PeleeNet is only 66% of the model size of MobileNet. Some of the key features of PeleeNet are:

- **Two-Way Dense Layer** Motivated by GoogLeNet [5], we use a 2-way dense layer to get different scales of receptive fields. One way of the layer uses a 3x3 kernel size. The other way of the layer uses two stacked 3x3 convolution to learn visual patterns for large objects. The structure is shown on Fig. 1,

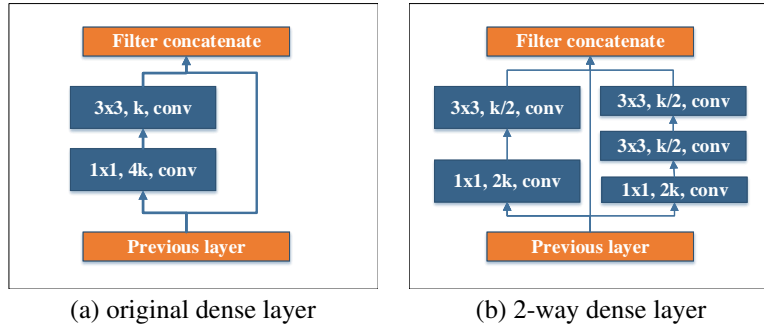

(a) original dense layer        (b) 2-way dense layer

Figure 1: Structure of 2-way dense layer

- **Stem Block** Motivated by Inception-v4 [10] and DSOD [11], we design a cost efficient stem block before the first dense layer. The structure of stem block is shown on Fig. 2. This stem block can effectively improve the feature expression ability without adding computational cost too much - better than other more expensive methods, e.g., increasing channels of the first convolution layer or increasing growth rate.

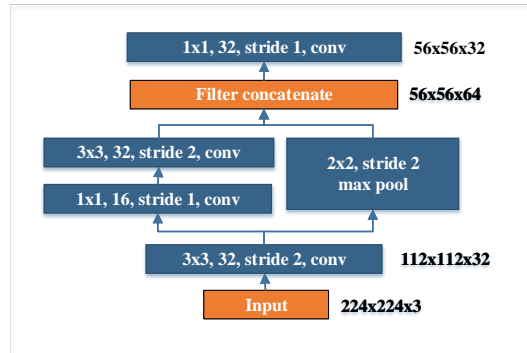

Figure 2: Structure of stem block

- **Dynamic Number of Channels in Bottleneck Layer** Another highlight is that the number of channels in the bottleneck layer varies according to the input shape instead of fixed 4 times of growth rate used in the original DenseNet. In DenseNet, we observe that for the first several dense layers, the number of bottleneck channels is much larger than the number of its input channels, which means that for these layers, bottleneck layer increases the computational cost instead of reducing the cost. To maintain the consistency of the architecture, we still add the bottleneck layer to all dense layers, but the number is dynamically adjusted according to the input shape, to ensure that the number of channels does not exceed the input channels. Compared to the original DenseNet structure, our experiments show that this method can save up to 28.5% of the computational cost with a small impact on accuracy. (Fig. 3)

- **Transition Layer without Compression** Our experiments show that the compression factor proposed by DenseNet hurts the feature expression. We always keep the number of output channels the same as the number of input channels in transition layers.

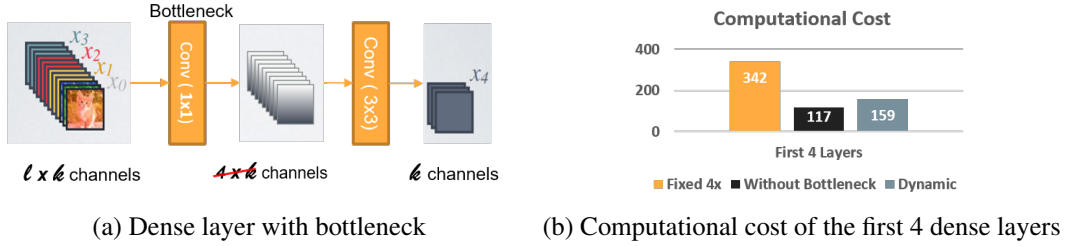

(a) Dense layer with bottleneck      (b) Computational cost of the first 4 dense layers

Figure 3: Dynamic number of channels in bottleneck layer

- **Composite Function** To improve actual speed, we use the conventional wisdom of "post-activation" (Convolution - Batch Normalization [12] - Relu) as our composite function instead of pre-activation used in DenseNet. For post-activation, all batch normalization layers can be merged with convolution layer at the inference stage, which can accelerate the speed greatly. To compensate for the negative impact on accuracy caused by this change, we use a shallow and wide network structure. We also add a 1x1 convolution layer after the last dense block to get the stronger representational abilities.

**We optimize the network architecture of Single Shot MultiBox Detector (SSD) [13] for speed acceleration and then combine it with PeleeNet.** Our proposed system, named Pelee, achieves 76.4% mAP on PASCAL VOC [14] 2007 and 22.4 mAP on COCO. It outperforms YOLOv2 [15] in terms of accuracy, speed and model size. The major enhancements proposed to balance speed and accuracy are:

- **Feature Map Selection** We build object detection network in a way different from the original SSD with a carefully selected set of 5 scale feature maps (19 x 19, 10 x 10, 5 x 5, 3 x 3, and 1 x 1). To reduce computational cost, we do not use 38 x 38 feature map.

- **Residual Prediction Block** We follow the design ideas proposed by [16] that encourage features to be passed along the feature extraction network. For each feature map used for detection, we build a residual [17] block (ResBlock) before conducting prediction. The structure of ResBlock is shown on Fig. 4

- **Small Convolutional Kernel for Prediction** Residual prediction block makes it possible for us to apply 1x1 convolutional kernels to predict category scores and box offsets. Our experiments show that the accuracy of the model using 1x1 kernels is almost the same as that of the model using 3x3 kernels. However, 1x1 kernels reduce the computational cost by 21.5%.

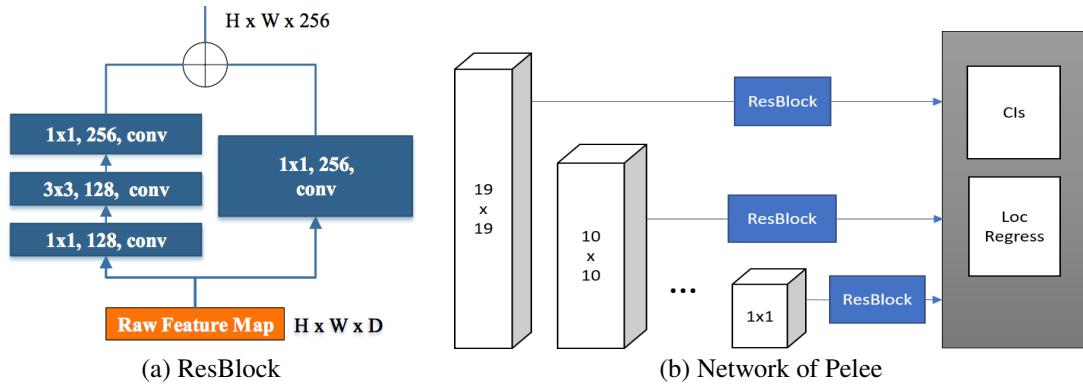

(a) ResBlock      (b) Network of Pelee

Figure 4: Residual prediction block

**We provide a benchmark test** for different efficient classification models and different one-stage object detection methods on NVIDIA TX2 embedded platform and iPhone 8.

# 2 PeleeNet: An Efficient Feature Extraction Network

## 2.1 Architecture

The architecture of our proposed PeleeNet is shown as follows in Table 1. The entire network consists of a stem block and four stages of feature extractor. Except the last stage, the last layer in each stage is average pooling layer with stride 2. A four-stage structure is a commonly used structure in the large model design. ShuffleNet [2] uses a three stage structure and shrinks the feature map size at the beginning of each stage. Although this can effectively reduce computational cost, we argue that early stage features are very important for vision tasks, and that premature reducing the feature map size can impair representational abilities. Therefore, we still maintain a four-stage structure. The number of layers in the first two stages are specifically controlled to an acceptable range.

Table 1: Overview of PeleeNet architecture

| Stage | | Layer | Output Shape |
|---|---|---|---|
| | Input | | 224 x 224 x 3 |
| Stage 0 | Stem Block | | 56 x 56 x 32 |
| Stage 1 | Dense Block | DenseLayer **x 3** | 28 x 28 x 128 |
| | Transition Layer | 1 x 1 conv, stride 1 | |
| | | 2 x 2 average pool, stride 2 | |
| Stage 2 | Dense Block | DenseLayer **x 4** | 14 x 14 x 256 |
| | Transition Layer | 1 x 1 conv, stride 1 | |
| | | 2 x 2 average pool, stride 2 | |
| Stage 3 | Dense Block | DenseLayer **x 8** | 7 x 7 x 512 |
| | Transition Layer | 1 x 1 conv, stride 1 | |
| | | 2 x 2 average pool, stride 2 | |
| Stage 4 | Dense Block | DenseLayer **x 6** | 7 x 7 x 704 |
| | Transition Layer | 1 x 1 conv, stride 1 | |
| Classification Layer | | 7 x 7 global average pool | 1 x 1 x 704 |
| | | 1000D fully-connecte,softmax | |

## 2.2 Ablation Study

### 2.2.1 Dataset

We build a customized Stanford Dogs dataset for ablation study. Stanford Dogs [8] dataset contains images of 120 breeds of dogs from around the world. This dataset has been built using images and annotation from ImageNet for the task of fine-grained image classification. We believe the dataset used for this kind of task is complicated enough to evaluate the performance of the network architecture. However, there are only 14,580 training images, with about 120 images per class, in the original Stanford Dogs dataset, which is not large enough to train the model from scratch. Instead of using the original Stanford Dogs, we build a subset of ILSVRC 2012 according to the ImageNet wnid used in Stanford Dogs. Both training data and validation data are exactly copied from the ILSVRC 2012 dataset. In the following chapters, the term of Stanford Dogs means this subset of ILSVRC 2012 instead of the original one. Contents of this dataset:

- Number of categories: 120
- Number of training images: 150,466
- Number of validation images: 6,000

### 2.2.2 Effects of Various Design Choices on the Performance

We build a DenseNet-like network called DenseNet-41 as our baseline model. There are two differences between this model and the original DenseNet. The first one is the parameters of the first

conv layer. There are 24 channels on the first conv layer instead of 64, the kernel size is changed from 7 x 7 to 3 x 3 as well. The second one is that the number of layers in each dense block is adjusted to meet the computational budget.

All our models in this section are trained by PyTorch with mini-batch size 256 for 120 epochs. We follow most of the training settings and hyper-parameters used in ResNet on ILSVRC 2012. Table 2 shows the effects of various design choices on the performance. We can see that, after combining all these design choices, PeleeNet achieves 79.25% accuracy on Stanford Dogs, which is higher in accuracy by 4.23% than DenseNet-41 at less computational cost.

Table 2: Effects of various design choices and components on performance

|  | From DenseNet-41 to PeleeNet | | | | | |
|---|---|---|---|---|---|---|
| Transition layer without compression | ✓ | ✓ | ✓ | ✓ | ✓ | ✓ |
| Post-activation |  | ✓ |  |  | ✓ | ✓ |
| Dynamic bottleneck channels |  |  | ✓ | ✓ | ✓ | ✓ |
| Stem Block |  |  |  | ✓ | ✓ | ✓ |
| Two-way dense layer |  |  |  |  | ✓ | ✓ |
| Go deeper (add 3 extra dense layers) |  |  |  |  |  | ✓ |
| Top 1 accuracy | **75.02** | 76.1 | 75.2 | 75.8 | 76.8 | 78.8 | **79.25** |

## 2.3 Results on ImageNet ILSVRC 2012

Our PeleeNet is trained by PyTorch with mini-batch size 512 on two GPUs. The model is trained with a cosine learning rate annealing schedule, similar to what is used by [18] and [19]. The initial learning rate is set to 0.18 and the total amount of epochs is 120. We then fine tune the model with the initial learning rate of 5e-3 for 20 epochs. Other hyper-parameters are the same as the one used on Stanford Dogs dataset.

**Cosine Learning Rate Annealing** means that the learning rate decays with a cosine shape (the learning rate of epoch $t$ ($t <= 120$) set to $0.5 * lr * (\cos(\pi * t/120) + 1)$).

As can be seen from Table 3, PeleeNet achieves a higher accuracy than that of MobileNet and ShuffleNet at no more than 66% model size and the lower computational cost. The model size of PeleeNet is only 1/49 of VGG16.

Table 3: Results on ImageNet ILSVRC 2012

| Model | Computational Cost (FLOPs) | Model Size (Parameters) | Accuracy (%) | |
|---|---|---|---|---|
|  |  |  | Top-1 | Top-5 |
| VGG16 | 15,346 M | 138 M | 71.5 | 89.8 |
| 1.0 MobileNet | 569 M | 4.24 M | 70.6 | 89.5 |
| ShuffleNet 2x (g = 3) | 524 M | 5.2 M | 70.9 | - |
| NASNet-A | 564 M | 5.3 M | 74.0 | 91.6 |
| **PeleeNet (ours)** | **508 M** | **2.8 M** | **72.1** | **90.6** |

## 2.4 Speed on Real Devices

Counting FLOPs (the number of multiply-accumulates) is widely used to measure the computational cost. However, it cannot replace the speed test on real devices, considering that there are many other factors that may influence the actual time cost, e.g. caching, I/O, hardware optimization etc,. This section evaluates the performance of efficient models on iPhone 8 and NVIDIA TX2 embedded platform. The speed is calculated by the average time of processing 100 pictures with 1 batch size. We run 100 picture processing for 10 times separately and average the time.

As can be seen in Table 4 PeleeNet is much faster than MoibleNet and MobileNetV2 on TX2. Although MobileNetV2 achieves a high accuracy with 300 FLOPs, the actual speed of the model is slower than that of MobileNet with 569 FLOPs.

Using half precision float point (FP16) instead of single precision float point (FP32) is a widely used method to accelerate deep learning inference. As can be seen in Figure 5, PeleeNet runs 1.8 times faster in FP16 mode than in FP32 mode. In contrast, the network that is built with depthwise separable convolution is hard to benefit from the TX2 half-precision (FP16) inference engine. The speed of MobileNet and MobileNetV2 running in FP16 mode is almost the same as the ones running in FP32 mode.

On iPhone 8, PeleeNet is slower than MobileNet for the small input dimension but is faster than MobileNet for the large input dimension. There are two possible reasons for the unfavorable result on iPhone. The first reason is related to CoreML which is built on Apple's Metal API. Metal is a 3D graphics API and is not originally designed for CNNs. It can only hold 4 channels of data (originally used to hold RGBA data). The high-level API has to slice the channel by 4 and caches the result of each slice. The separable convolution can benefit more from this mechanism than the conventional convolution. The second reason is the architecture of PeleeNet. PeleeNet is built in a multi-branch and narrow channel style with 113 convolution layers. Our original design is misled by the FLOPs count and involves unnecessary complexity.

Table 4: **Speed on NVIDIA TX2** (The larger the better) The benchmark tool is built with NVIDIA TensorRT4.0 library.

| Model | Top-1 Accuracy on ILSVRC2012 @224x224 | FLOPs @224x224 | Speed (images per second) | | |
|---|---|---|---|---|---|
| | | | Input Dimension | | |
| | | | 224x224 | 320x320 | 640x640 |
| 1.0 MobileNet | 70.6 | 569 M | 136.2 | 75.7 | 22.4 |
| 1.0 MobileNetV2 | 72.0 | 300 M | 123.1 | 68.8 | 21.6 |
| ShuffleNet 2x (g = 3) | 70.9 | 524 M | 110 | 65.3 | 19.8 |
| **PeleeNet (ours)** | **72.1** | 508 M | **240.3** | **129.1** | **37.2** |

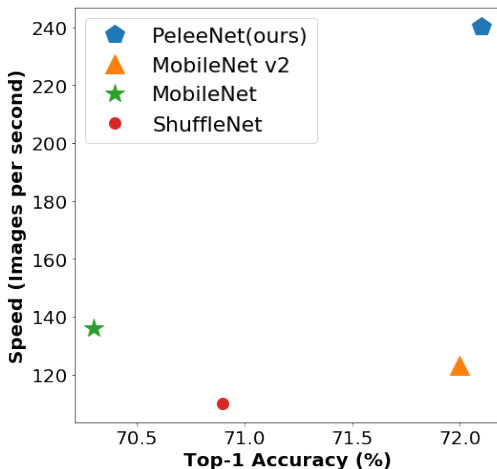

(a) Speed and accuracy on FP16 mode

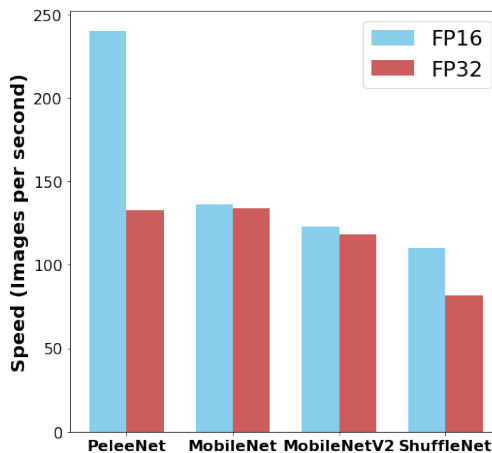

(b) FP32 vs FP16 by 224x224 dimension

Figure 5: Speed on NVIDIA TX2

Table 5: **Speed on iPhone 8** (The larger the better) The benchmark tool is built with CoreML library.

| Model | Top-1 Accuracy on ILSVRC2012 @224x224 | FLOPs @224x224 | Speed (images per second) | |
|---|---|---|---|---|
| | | | Input Dimension | |
| | | | 224x224 | 320x320 |
| 1.0 MobileNet | 70.6 | 569 M | **27.7** | 20.3 |
| **PeleeNet (ours)** | **72.1** | 508 M | 26.1 | **22.8** |

# 3 Pelee: A Real-Time Object Detection System

## 3.1 Overview

This section introduces our object detection system and the optimization for SSD. The main purpose of our optimization is to improve the speed with acceptable accuracy. Except for our efficient feature extraction network proposed in last section, we also build the object detection network in a way different from the original SSD with a carefully selected set of 5 scale feature maps. In the meantime, for each feature map used for detection, we build a residual block before conducting prediction (Fig. 4). We also use small convolutional kernels to predict object categories and bounding box locations to reduce computational cost. In addition, we use quite different training hyperparameters. Although these contributions may seem small independently, we note that the final system achieves 70.9% mAP on PASCAL VOC2007 and 22.4 mAP on MS COCO dataset. The result on COCO outperforms YOLOv2 in consideration of a higher precision, 13.6 times lower computational cost and 11.3 times smaller model size.

There are 5 scales of feature maps used in our system for prediction: 19 x 19, 10 x 10, 5 x 5, 3 x 3, and 1 x 1. We do not use 38 x 38 feature map layer to ensure a balance able to be reached between speed and accuracy. The 19x19 feature map is combined to two different scales of default boxes and each of the other 4 feature maps is combined to one scale of default box. Huang et al. [6] also do not use 38 x 38 scale feature map when combining SSD with MobileNet. However, they add another 2 x 2 feature map to keep 6 scales of feature map used for prediction, which is different from our solution.

Table 6: Scale of feature map and default box

| Model | Scale of Feature Map : Scale of Default Box | | | | | |
|---|---|---|---|---|---|---|
| Original SSD | 38x38:30 | 19x19:60 | 10x10:110 | 5x5:162 | 3x3:213 | 1x1:264 |
| SSD + MobileNet [6] | 19x19:60 | 10x10:105 | 5x5:150 | 3x3:195 | 2x2:240 | 1x1:285 |
| **Pelee (ours)** | 19x19: 30.4 & 60.8 | | 10x10:112.5 | 5x5:164.2 | 3x3:215.8 | 1x1:267.4 |

## 3.2 Results on VOC 2007

Our object detection system is based on the source code of SSD[3] and is trained with Caffe [20]. The batch size is set to 32. The learning rate is set to 0.005 initially, then it decreased by a factor of 10 at 80k and 100k iterations,respectively. The total iterations are 120K.

### 3.2.1 Effects of Various Design Choices

Table 7 shows the effects of our design choices on performance. We can see that residual prediction block can effectively improve the accuracy. The model with residual prediction block achieves a higher accuracy by 2.2% than the model without residual prediction block. The accuracy of the model using 1x1 kernels for prediction is almost same as the one of the model using 3x3 kernels. However, 1x1 kernels reduce the computational cost by 21.5% and the model size by 33.9%.

Table 7: Effects of various design choices on performance

| 38x38 Feature | ResBlock | Kernel Size for Prediction | FLOPs | Model Size (Parameters) | mAP (%) |
|:---:|:---:|:---:|:---:|:---:|:---:|
| ✓ | ✗ | 3x3 | 1,670 M | 5.69 M | 69.3 |
| ✗ | ✗ | 3x3 | 1,340 M | 5.63 M | 68.6 |
| ✗ | ✓ | 3x3 | 1,470 M | 7.27 M | 70.8 |
| ✗ | ✓ | 1x1 | 1,210 M | 5.43 M | **70.9** |

### 3.2.2 Comparison with Other Frameworks

As can be seen from Table 8, the accuracy of Pelee is higher than that of TinyYOLOv2 by 13.8% and higher than that of SSD+MobileNet [6] by 2.9%. It is even higher than that of YOLOv2-288 at only 14.5% of the computational cost of YOLOv2-288. Pelee achieves 76.4% mAP when we take the model trained on COCO trainval35k as described in Section 3.3 and fine-tuning it on the 07+12 dataset.

Table 8: **Results on PASCAL VOC 2007.** Data: "07+12": union of VOC2007 and VOC2012 trainval. "07+12+COCO": first train on COCO trainval35k then fine-tune on 07+12

| Model | Input Dimension | FLOPs | Model Size (Parameters) | Data | mAP (%) |
|:---:|:---:|:---:|:---:|:---:|:---:|
| YOLOv2 | 288x288 | 8,360 M | 67.13 M | 07+12 | 69.0 |
| Tiny-YOLOv2 | 416x416 | 3,490 M | 15.86 M | 07+12 | 57.1 |
| SSD+MobileNet | 300x300 | 1,150 M | 5.77 M | 07+12 | 68 |
| **Pelee (ours)** | 304x304 | 1,210 M | 5.43 M | 07+12 | **70.9** |
| SSD+MobileNet | 300x300 | 1,150 M | 5.77 M | 07+12+COCO | 72.7 |
| **Pelee (ours)** | 304x304 | 1,210 M | 5.43 M | 07+12+COCO | **76.4** |

### 3.2.3 Speed on Real Devices

We then evaluate the actual inference speed of Pelee on real devices. The speed are calculated by the average time of 100 images processed by the benchmark tool. This time includes the image pre-processing time, but it does not include the time of the post-processing part (decoding the bounding-boxes and performing non-maximum suppression). Usually, post-processing is done on the CPU, which can be executed asynchronously with the other parts that are executed on mobile GPU. Hence, the actual speed should be very close to our test result.

Although residual prediction block used in Pelee increases the computational cost, Pelee still runs faster than SSD+MobileNet on iPhone and on TX2 in FP32 mode. As can be seen from Table 9, Pelee has a greater speed advantage compared to SSD+MobileNet and SSDLite+MobileNetV2 in FP16 mode.

Table 9: Speed on real devices

| Model | Input Dimension | FLOPs | Speed (FPS) | | |
|:---:|:---:|:---:|:---:|:---:|:---:|
| | | | iPhone 8 | TX2 (FP16) | TX2 (FP32) |
| SSD+MobileNet | 300x300 | 1,200 M | 22.8 | 82 | 73 |
| SSDLite+MobileNetV2 | 320x320 | 805 M | - | 62 | 60 |
| **Pelee (ours)** | 304x304 | 1,290 M | **23.6** | **125** | **77** |

## 3.3 Results on COCO

We further validate Pelee on the COCO dataset. The models are trained on the COCO train+val dataset excluding 5000 minival images and evaluated on the test-dev2015 set. The batch size is set

to 128. We first train the model with the learning rate of $10^{-2}$ for 70k iterations, and then continue training for 10k iterations with $10^{-3}$ and 20k iterations with $10^{-4}$.

Table 10 shows the results on test-dev2015. Pelee is not only more accurate than SSD+MobileNet [6], but also more accurate than YOLOv2 [15] in both mAP@[0.5:0.95] and mAP@0.75. Meanwhile, Pelee is 3.7 times faster in speed and 11.3 times smaller in model size than YOLOv2.

Table 10: Results on COCO test-dev2015

| Model | Input Dimension | Speed on TX2 (FPS) | Model Size (Parameters) | Avg. Precision (%), IoU: | | |
|---|---|---|---|---|---|---|
| | | | | 0.5:0.95 | 0.5 | 0.75 |
| Original SSD | 300x300 | - | 34.30 M | 25.1 | 43.1 | 25.8 |
| YOLOv2 | 416x416 | 32.2 | 67.43 M | 21.6 | 44.0 | 19.2 |
| YOLOv3 | 320x320 | 21.5 | 62.3 M | - | 51.5 | - |
| YOLOv3-Tiny | 416x416 | 105 | 12.3 M | - | 33.1 | - |
| SSD+MobileNet | 300x300 | 80 | 6.80 M | 18.8 | - | - |
| SSDlite + MobileNet v2 | 320x320 | 61 | 4.3 M | 22 | - | - |
| **Pelee (ours)** | 304x304 | 120 | 5.98 M | **22.4** | **38.3** | **22.9** |

# 4  Conclusion

Depthwise separable convolution is not the only way to build an efficient model. Instead of using depthwise separable convolution, our proposed PeleeNet and Pelee are built with conventional convolution and have achieved compelling results on ILSVRC 2012, VOC 2007 and COCO.

By combining efficient architecture design with mobile GPU and hardware-specified optimized runtime libraries, we are able to perform real-time prediction for image classification and object detection tasks on mobile devices. For example, Pelee, our proposed object detection system, can run 23.6 FPS on iPhone 8 and 125 FPS on NVIDIA TX2 with high accuracy.

## Footnotes

*Contact author

[2]The code and models are available at: https://github.com/Robert-JunWang/Pelee

[3]https://github.com/weiliu89/caffe/tree/ssd

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
