[Reviews · NeurIPS 2018]

Reviewer 1



Pelee is a variation on DenseNet which is a CNN network architecture that has many shortcut connections between layers (within a DenseNet block all layers are connected to all previous layers in the block with shortcut connections) The authors make a number of modifications to the original DenseNet and combine it with a modified version of SSD (Sinlge Shot Detector). They make many design choices to optimize speed in both DenseNet and SSD and find that they are able to maintain accuracy; thus providing what seems to be a very practical network for use on low power devices (eg. iPhones) at reasonable frame rates. The modifications to DensNet and SSD and justifications are described in sufficient detail. The modifications are not blindingly novel, but are well thought out and effective as a whole, which is the point. The comparisons with other state of the art optimized CNN (multi-object localization) architectures appear reasonable. YOLOv2(-228) and MobileNet-SSD are the right models to compare to. This is good work, with good results, the code is online and I am sure it will be useful to many. Note: There are a few references to YOLOv2 without saying YOLOv2-228. I assume all comparisons are made to the 228x228 pixel input version of YOLO, but this isn't clear from the start. YOLOv2 (high res) is better for accuracy without considering speed or size.

Reviewer 2



The paper address the problem of accurate object detection on mobile device which an important problem has not been solved. Current accurate detectors rely on large and deep networks which only be inferred on a GPU. To address this problem, it proposes a SSD based detection method based on a new network termed as Pelee. Pros: + The Pelee detector achieves 76.4% mAP on PASCAL VOC2007 and 22.4 mAP on MS COCO dataset at the speed of 17.1 FPS on iPhone 6s and 23.6 FPS on iPhone 8. The accuracies are getting very close to the SSD detectors with VGGNets. The result on COCO outperforms YOLOv2 in consideration of a higher precision, 13.6 times lower computational cost and 11.3 times smaller model size. The results are very strong. + Some new designs which are different and effective that make Pelee works. (1) Two-way dense layer motivated by GoogLeNet. (2) A cost-efficient stem block before the first dense layer. (3) Dynamic number of channels in bottleneck layer: the number is dynamically adjusted according to the input shape, to ensure that the number of channels does not exceed the input channels. (4) Transition layer without compression: it uses the conventional wisdom of “post-activation” (Convolution - Batch Normalization [11] - Relu) as the composite function. (5) Feature map selection, selected set of 5 scale feature maps (19 x 19, 10 x 10, 5 x 5, 3 x 3, and 1 x 1). To reduce computational cost, it does not use 38 x 38 feature map. (6) Residual SSD prediction block. (7) Small convolutional kernel for location and confidence score prediction. Cons: - The technical contributions are not significant enough, since it is in the existing SSD detection framework and the network design are mainly based on experience. There is no theoretical finding. - It lacks comparisons to the state-of-the-art mobile deep detectors, such as SSD with MobileNetV2 & NasNet. - The efficient implementation of SSD on iOS cannot be regarded as a contribution since it simply based on the CoreML library. Some minor issues: - The format and the presentation of the paper are not good. Tab 2&5 are not not well-aligned which make people hard to read. - Some grammar mistakes: e.g., the “innovate” connectivity pattern. - Why the method is termed as “Pelee”?

Reviewer 3



The authors proposes a new network backbone and applied it to improve object detection model speed and accuracy under constrained computation budget. The backbone is modified from densenet and detection architecture modified from SSD. The authors compare their backbone model in their own dataset and ImageNet, and detection model in PASCAL VOC and COCO. Strengths: - The proposed method is described in detail and evaluated at component level and end-to-end. - Proposed detection model improves AP by ~4% compared with mobilenetSSD, which is significant. - Authors also compared the runtime in actual device, which is more convincing than just counting FLOPs. Weaknesses: - Although authors have proposed a set of modifications to their baseline DenseNet + SSD model, the novelty on each modification is limited. - Section 2.2.3. The experiment comparing PeleeNet with other network architecture is flawed. The author says "we can evaluate a model pre-trained on ILSVRC 2012 on this dataset", but the whole ImageNet has 1000 object classes but Stanford Dogs dataset only has 120 dog classes. Although some features can be shared in lower layers, comparing a model trained for a variety of categories with a model for dogs only is not fair. Also Table 3, PeleeNet shows big improvement on stanford dogs (~7% top-1 gain over mobilenet) but only 0.6% on the full Imagenet also suggests this experiment is not solid. Some additional comments: - It makes a lot of sense separable conv has an advantage in FLOP counts but doesn't run as fast in actual devices, due to bad memory access pattern. However the final Pelee model (which uses conventional conv) has similar MACs with mobilenet SSD and similar runtime too. Additional analysis / profiling could make this paper stronger. - Section 1, paragraph 3: "the layer uses a small kernel size (3x3), which is good enough to capture small-size objects": 3x3 is a standard practice for most modern NNs. It is arguable 3x3 is "good enough". Also smaller kernel is not very relevant with small objects? What matters more is better spatial resolution in conv feature maps, which is usually decided by the stride of conv kernels rather than its size. - Section 1, paragraph "Composite Function": "post-activation" is referring to BN before ReLU, this name is a bit confusing... better call it "BN before activation" / "BN after activation"? - Same paragraph: experiment (Table 2) does show BN before ReLU hurt performance. The authors argue BN can be folded into conv this way, but BN after ReLU can be folded into next Conv op, because all activation in the same conv feature map shares the same mean / var / scale / offset in BN implementation? - The proposed architecture drops 38x38 feature map which would hurt on small objects? Usually small objects would cost more budget to capture and they don't give good improvement on overall mAP. It would be more convincing to analyze how the proposed model works for small / medium / large size objects. - [Minor] Section 2.1. The proposed model uses avg pool with stride 2. Conv with stride 2 could potentially be a good alternative? - [Minor] Section 3.1: SSD feature map used in this paper is very similar to [5]. Dropping 2x2 map has limited impact on speed / accuracy. Meanwhile, Mobilenet V1 / V2 + SSD lite (https://arxiv.org/pdf/1801.04381.pdf) presents similar performance but with smaller size / FLOPs. However it is reasonable to assume this paper to be concurrent work.